



**The effects of Hurricane Harvey on Texas coastal zone chemistry**
Piers Chapman[1,2], Steven F. DiMarco[1,2], Anthony H. Knap[1,2], Antonietta Quigg[1,3], Nan D.
Walker[4]
1. Department of Oceanography, Texas A&M University, College Station, TX 77843
2. Geochemical and Environmental Research Group, Texas A&M University, College
Station, TX 77843
3. Department of Marine Biology, Texas A&M University, Galveston, TX 77553
4. Department of Oceanography and Coastal Sciences, Louisiana State University, Baton Rouge,
LA, 70803
*Correspondence to*: Piers Chapman (piers.chapman@tamu.edu)
**Abstract**
Hurricane Harvey deposited over 90 billion cubic meters of rainwater over central Texas, USA,
during late August/early September 2017. During four cruises (June, August, September and
November 2017) we observed changes in hydrography, nutrient and oxygen concentrations in
Texas coastal waters. Despite intense terrestrial runoff, nutrient supply to the coastal ocean was
transient, with little phytoplankton growth observed and no hypoxia. Observations suggest this
was probably related to the retention of nutrients in the coastal bays, rapid uptake by
phytoplankton of nutrients washed out of the bays, as well as dilution by the sheer volume of
rainwater, and the lack of significant carbon reserves in the sediments, despite the imposition of
a strong pycnocline. By the November cruise conditions had apparently returned to normal and
no long-term effects were observed.
**Keywords**
Hurricane Harvey, Texas coast, nutrients, oxygen, chlorophyll



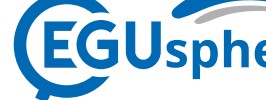

## 1. Introduction

The Gulf of Mexico is renowned for its hurricanes and tropical storms, and the Galveston hurricane of 1900, with an estimated total fatality of over 8,000, has the distinction of still being the worst natural disaster in the U.S. in recorded history (Larson, 1999). More recently, hurricanes such as Andrew (1992), Katrina, Rita and Wilma (2005), and Ike (2008) have ravaged the northern Gulf coast, causing extensive damage and large financial losses. 2017 was another active year in the Atlantic, with 10 hurricanes and 8 tropical cyclones and depressions, including the category 5 hurricanes Irma, which affected several Caribbean islands and reached Florida as a category 3 storm in late August/early September, and Maria, which devastated Dominica and Puerto Rico in September (see the archives at http://www.nhc.noaa.gov).

Hurricane Harvey developed in the Bay of Campeche, in the extreme southwestern part of the Gulf of Mexico, on 23 August, 2017, intensifying rapidly on August 24 over water with SST >30° C and an upper ocean heat content anomaly (measured by three ARGOS floats) that extended to ~45 m water depth (Trenberth et al., 2018). Harvey crossed the edge of the Texas shelf in the northwestern Gulf at 18.00 U.S. Central Time having intensified to category 3, and reached category 4 strength by midnight of August 25 with sustained wind speeds of 60 m/s (115 kt) and a minimum central pressure of 937 mbar (Blake and Zelinsky 2018). Rapid intensification of tropical cyclones over the shallow waters of the south Texas shelf has been reported previously and is believed to be related to periods when warm water occupies the whole water column. This prevents mixing of colder bottom water that can reduce the energy flux feeding the hurricane (Potter et al., 2019). The storm came ashore near Corpus Christi, TX on 26 August, and stalled over the TX coast, moving slowly to the northeast until August 31, after which it moved inland and dissipated over Kentucky (Fig. 1).

Harvey brought a storm surge of up to 3 m and torrential rain to the Texas coast, with more than 1200 mm (48 in) of rain being recorded at 18 stations during its passage. The heaviest rainfall was measured in Harris County, TX, at Nederland and Groves, near Houston, where over 1500 mm (60 in) fell (Blake and Zelinsky, 2018). Heavy rain (<500 mm) also affected Louisiana (Fig.1). This unprecedented rainfall, the highest ever recorded in the U.S. for a tropical cyclone, resulted in widespread flooding in Texas and Louisiana, more than 80 fatalities, and over $150





Fig. 1. Track of Hurricane Harvey and associated rainfall over the southern United States, August 24-September 4, 2017 (from Blake and Zelinsky, 2018). The numbers 1 and 2 denote the positions of Galveston Bay and Matagorda Bay respectively.

billion in economic damage (Emanuel, 2017; Balaguru et al., 2018). It is estimated that the total amount of water that fell as rain over Texas and Louisiana during Harvey's passage was between $92.7 \times 10^9$ m$^3$ (Fritz and Samenow, 2017), and $133 \times 10^9$ m$^3$ (DiMarco, unpublished), and over 200 mm of rain was recorded as far inland as Tennessee and Kentucky as the storm died down (Blake and Zelinski, 2018; Fig.1). In addition to the rain that fell on land, DiMarco (unpublished) has estimated that about another $44 \times 10^9$ m$^3$ fell over the ocean.

Galveston Bay collects the runoff from the Houston metropolitan region. Following the storm, the bay became a freshwater lake (Du et al., 2019; Steichen et al., 2020; Thyng et al., 2020) as it was flushed with about three to five times its volume of rainwater. U.S. Geological Survey (USGS) data (downloaded from https://waterdata.usgs.gov) show very rapid increases in flow rates in Texas rivers and streams following the storm's landfall. For instance, flows in the Colorado and Brazos Rivers (USGS stations 08162000 and 08111500 respectively; Figs S1a and S1b) increased from <2,000 cfs (~60 m$^3$/s) during most of August to over 90,000 cfs (>2,500 m$^3$/s) by the beginning of September, while flow in both the San Jacinto River (USGS station



08068090, Fig. S1c) and the Trinity River at Liberty (USGS station 08067000, Fig. S1d)
exceeded 100,000 cfs (~3,400 m³/s). Flow in the Trinity River, which is generally the major
source of fresh water to Galveston Bay, increased from 20,800 cfs (~600 m³/s) on August 27 to
100,000 cfs (~3,400 m³/s) on August 31. The gauge at this site was unfortunately not in
operation immediately prior to August 27 or after September 9, but during June flowrates were
typically 10,000 – 14,000 cfs (~300-420 m³/s). Such large changes in runoff are known to
produce major changes in estuaries and coastal waters (e.g., Ahn et al., 2005; Paerl et al., 2001,
2006; Mallin and Corbett, 2006; De Carlo et al., 2007; Zhang et al., 2009; Du et al., 2019; Thyng
et al., 2020). Runoff can add nutrients, heavy metals, oil and other organics, soil, and debris, all
of which can affect the local biota either positively (e.g., increasing local productivity through
nutrient input) or negatively (e.g., through salinity reduction, toxicity, smothering or reducing
biomass through eutrophication). Liu et al. (2019) and Steichen et al. (2020) reported changes in
the phytoplankton community within Galveston Bay as the salinity decreased and then increased
again.

Given the amount of rainwater released during the passage of the hurricane, it is not surprising
that there was massive runoff, including turbidity plumes that were visible well offshore (Fig.
S2). D'Sa et al. (2018) monitored large increases in terrestrial carbon (25.22 x 10⁶ kg) and
suspended sediments (314.7 x 10⁶ kg) entering Galveston Bay during the period 26 August-4
September. The plume off Galveston Bay on 31 August extended at least 55 km offshore (Du et
al., 2019), and surface water with a salinity of 15 was measured on 1 September at the Texas
Automated Buoy System (TABS) buoy F (28.84°N, 94.24°W; yellow diamond in Fig. S2),
where it is typically 31-32 (data from https://tabs.gerg.tamu.edu). Normal salinities did not return
until 8 September. Similar sediment plumes at the mouths of the Brazos and Guadalupe estuaries
can be seen in Fig. S2, and such plumes and lowered salinities have been reported from the
Lavaca-Colorado and Nueces-Corpus estuaries near Corpus Christi (Walker et al., 2021). It is
likely that other bays and estuaries along the Texas coast were similarly affected, as they were all
under the path of the hurricane.

We report here on data collected before and after the hurricane along the Texas coast between
Galveston and Padre Island, Texas. Two cruises were completed prior to the advent of Hurricane



Harvey as part of a separate project. Following the hurricane, we completed three more cruises,
occupying the same stations in September (twice) and November 2017. This paper reports on the
changes in the water column between the pre- and post-hurricane cruises as they relate to
stratification, nutrient supply and oxygen concentrations.

**2. Methods**

Pre-hurricane cruises on the R.V. *Manta* took place in June (12-16) and August (7-11) 2017,
while post-hurricane cruises were from 22-27 September, 29 September – 1 October, and 15-20
November on the R.V. *Point Sur*. The 27 September-1 October cruise only occupied the two
inshore stations on each line; all other cruises covered a standard grid of five lines of five
stations each (Fig. 2), together with supplemental *ad hoc* stations between lines and offshore in
the east of the region towards the Flower Gardens Banks National Marine Sanctuary, a shallow
reef system 120 km south of Galveston Bay near 27.92°N, 93.75°W. During the November
cruise, additional stations were added at the outer ends of the southernmost lines to ensure
sampling of Gulf of Mexico offshore surface water with salinity >35. Depths at the outer ends of
each line decreased from 95-110 m at stations 5 and 10 to 85 m at station 15, and 50 m at stations
20 and 25.

At each station, a full-depth CTD cast was made using a SeaBird 911 CTD fitted with a SBE-55
temperature sensor, SBE-3 conductivity sensor, SBE-45 pressure sensor, and a SBE-43 oxygen
probe. Additional sensors on the rosette package included a Chelsea Instruments Aqua3
fluorometer and a Biosperical/Licor PAR sensor.  Discrete samples were collected from a 6-
bottle rosette for salinity determinations ashore and for oxygen calibration by Winkler titration
on board ship. Nutrient samples were collected, filtered, frozen on board and analyzed ashore for
nitrate, nitrite, phosphate, silicate, and ammonia by standard autoanalyzer methods (WHPO
1994). Limits of detection are about 0.1 $\mu$mol/L for nitrate, silicate and ammonia, and 0.02
$\mu$mol/L for nitrite and phosphate. Local meteorological data were collected by the ship's system,
while surface water temperature and salinity data came from the ships' flow-through system.





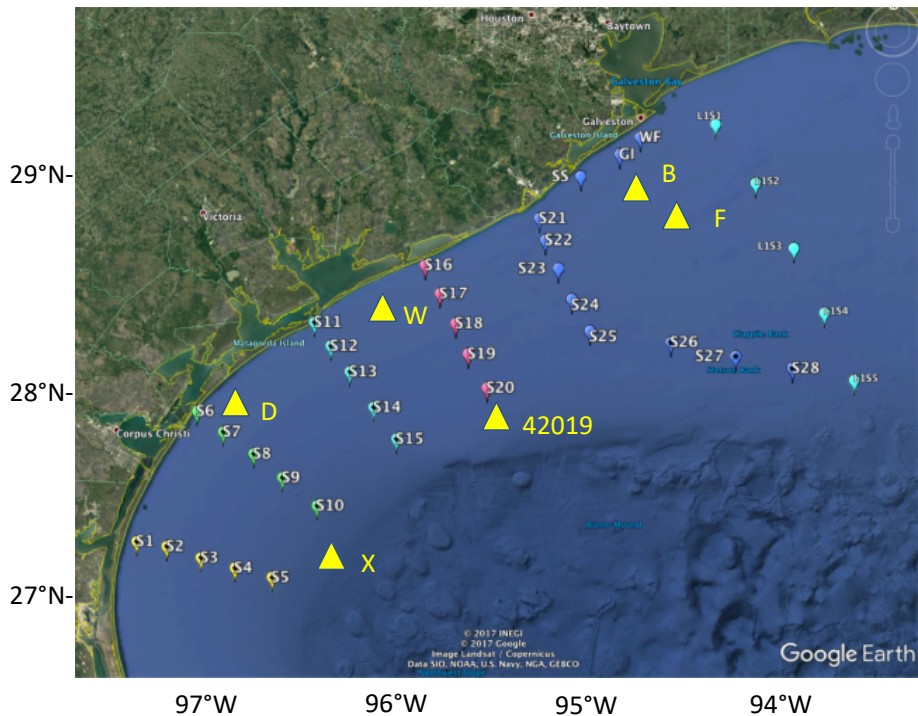

Fig. 2. Stations occupied during the four cruises. Only stations S1-S25 and the inshore stations GI, SS and WF were occupied during June and August. All stations shown were occupied in September (22-27) and November. Only the two inshore stations on each line were occupied during the second September cruise. Yellow triangles show positions of TABS moorings B, D, F, and W, and NOAA buoy 42019.

Wind and current data are available from the TABS moorings along the Texas coast (see Fig. 2 for positions and http://tabs.gerg.tamu.edu for the data archive). Buoy B (off Galveston) provided both wind and current data from before Harvey's landfall with a gap in the first half of August); buoys W (off Matagorda Bay) and D (off Corpus Christi) provided current data only. We have used additional wind data from TABS buoy X, which provided data until it failed on the morning of 25 September, and NOAA buoy 42019 (29.91°N, 95.34°W, obtained from the National Data Buoy Center at https://www.ndbc.noaa.gov).

Fluorometer data were obtained at each station sampled using a Chelsea Aqua 3 instrument on the rosette. Satellite imagery (Level 2 Ocean Color files) obtained by the Aqua-1 MODIS sensor downloaded from the NASA Goddard ocean color website (https://oceancolor.gsfc.nasa.gov)





were processed using the NASA SeaDAS software. In reality, the satellite-derived values may be
too high, due to the presence of CDOM after the storm (D'Sa et al., 2018), as the OC3 algorithm
provided by the SeaDAS software cannot discriminate between chlorophyll *a* and CDOM.

**3.  Results**
**3.1 Wind fields**
Wind data from all moorings (not shown) were typical of summer conditions in this part of the
Gulf of Mexico, being predominantly from the south with occasional reversals (Nowlin et al.,
1998). At TABS buoy B, wind velocities during June and July were generally 5-8 m/s and varied
between SSE and SSW. Following a gap in data from 31 July until 22 August, they remained in
this quadrant until the passage of the hurricane, although wind speeds increased from 3-4 m/s on
August 22 to 12 m/s on August 29 when they were from the north. Following the passage of the
hurricane, winds again were predominantly from the SE/SSE during September, with the
exception of two short-lived reversals on September 5 and 10-12. Wind speeds during these
reversals were around 4-7 m/s.

Further south and offshore, at TABS mooring X and NOAA mooring 42019, weak northerly
winds (generally <4 m/s) were experienced from 6-8 June, with a second northerly spell from 20-
22 June, when speeds reached 10 m/s and mooring X and 15 m/s at 42019. After this second
frontal system moved through, winds reverted to SE/SSE at both moorings until the passage of
Hurricane Harvey at the end of August. During September, at mooring 42019, winds were
primarily from the NNE/ENE at 4-10 m/s until the 12[th], and again from the 27[th], with SE or
easterly winds of 3-7 m/s from September 14-26. Maximum sustained wind speeds recorded
during the hurricane at this mooring were 17 m/s, with gusts to 22.6 m/s. During October, there
were two northerly/westerly wind events, on the 16[th], when winds reached speeds of 15m/s, and a
sustained event from 25-28 October, again with speeds <15 m/s. Northerly winds continued
during November, with sustained winds of 12-14 m/s during the periods 8-11, 18-20, and 22-24.

**3.2 Water movement**
Water movement over the Texas shelf is typically downcoast (towards the southwest) in non-
summer months and upcoast (towards the northeast) in summer, with currents following the wind



(Cochrane and Kelly, 1986; Walker, 2005). Upcoast winds and currents promote upwelling and
act to retain water from the Mississippi-Atchafalaya system on the east Texas-Louisiana shelf
(Hetland and DiMarco, 2008), while downcoast flow is downwelling-favorable and can reduce
local stratification. During June 2017, currents at Buoy D (27.96° N, 96.84° W) were essentially
downcoast from prior to the cruise until June 15, when they switched to upcoast until June 20,
after which they flowed downcoast again (Fig. 3a). The current reversal took place slightly later
(June 17) at Buoys B (28.98° N, 94.90° W) and W (28.35° N, 96.02° W), but the return to
downcoast flow again occurred on 20 June at both sites (Fig. 3a). These three moorings are all
situated close to the coast in water depths of 20 +/- 2 m.

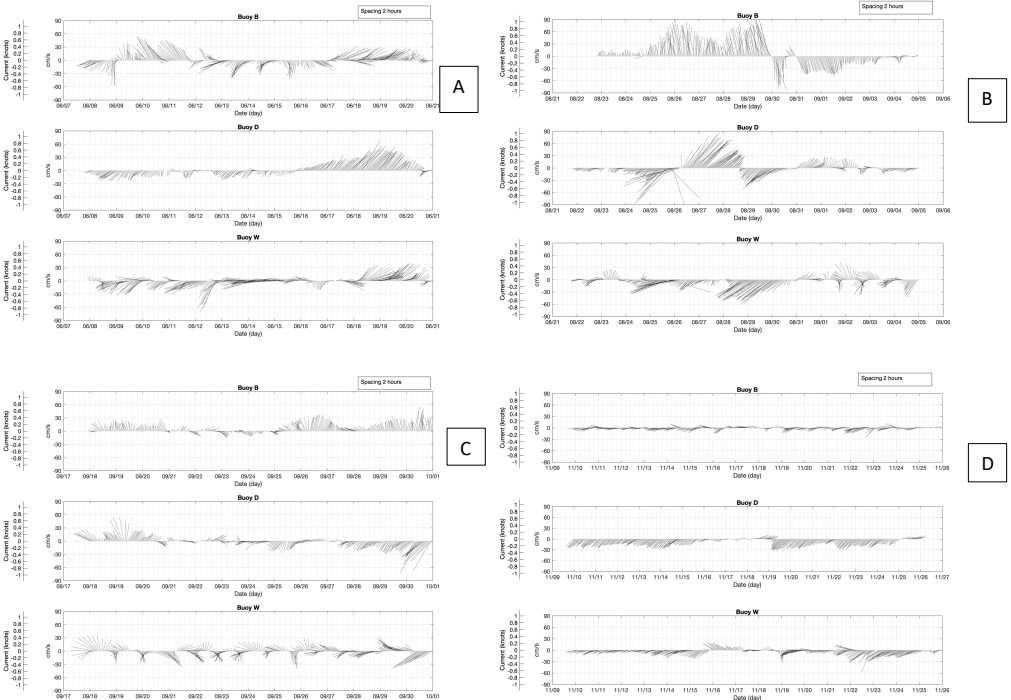


Fig. 3. Current vectors at TABS buoys B, D and W during (A) the June cruise, (B) the period of the hurricane
(August), and the cruises in September (C) and November (D).

Upcoast currents prevailed at sites W and D during the August cruise (Fig. 3), although currents
were downcoast from about August 8-10 at W and 9-11 at site D (not shown). Buoy B did not





record current speeds during this period, but was back in service immediately before the
hurricane arrived. During the passage of the hurricane, the southernmost mooring (buoy D)
recorded strong currents of > 1 m/s which changed from downcoast to upcoast and back to
downcoast again as the storm moved towards the northeast (Fig. 3b). Buoy W recorded
continuous downcoast currents during the period of the hurricane, while buoy B showed strong
onshore currents (<1.0 m/s) until August 30, when currents reversed to offshore at < 80 cm/s.
Following the hurricane, coastal currents were considerably weaker at all three sites in
September and November. During the September cruise there were a number of current
reversals, especially at buoy W, although velocities were generally <30 cm/s (Fig. 3c). By
November, current velocities decreased still further and the expected flow towards the west was
reinstated (Fig. 3d).

**3.3 Temperature, precipitation and salinity**
Temperatures measured during the cruises (not shown) showed well-mixed or weakly stratified
water inshore in June and August with surface-bottom differences of less than 2° at the two
inshore stations on each line. Further offshore, bottom temperatures decreased with depth but
there remained a well-mixed surface layer of 15-25m thickness. Following the hurricane,
however, the mixed layer extended further offshore, including the third station along each line in
September and almost all stations in November, when isothermal water was found as deep as
80m in some instances, and bottom temperatures were often warmer than at the surface.

Surface temperatures across the region increased from about 28.5 °C during the June cruise to
over 30 °C in August (Trenberth et al., 2018). As the hurricane passed through the region,
temperatures measured at the buoys, including at NBDC buoy 42019 (27.91° N, 95.34° W),
decreased to a minimum of about 27.5 °C, but recovered to 28.5-29 °C by the time of the
September cruises. By November, temperatures had decreased to 21-22 °C, 22-23 °C and 23-
23.5 °C at buoys B, W and D respectively. NBDC buoy 42019, which is further offshore than the
TABS moorings in 82 m of water, registered temperatures of between 25.4 and 26.0 °C during
this period.



Table 1. Precipitation rates (cm) for sites in central Texas from May-September 2017 compared with the long-term mean (italics). Data downloaded from https://www.srcc.tamu.edu/climate_data_portal/?product=precip_summary (accessed 7.07.2021).

| | May | June | July | Aug | Sept |
|---|---|---|---|---|---|
| Austin International airport | 7.59 | 6.17 | 2.69 | 32.99 | 9.68 |
| | *11.86* | *8.28* | *4.65* | *6.20* | *8.46* |
| Corpus Christi airport | 8.18 | 4.90 | 3.22 | 14.98 | 3.71 |
| | *8.51* | *8.00* | *5.97* | *7.87* | *13.41* |
| Houston Hobby airport | 6.81 | 13.20 | 7.92 | 98.73 | 9.52 |
| | *12.80* | *13.84* | *11.40* | *11.81* | *13.13* |
| Houston Intercontinental airport | 6.12 | 18.26 | 15.98 | 99.34 | 3.12 |
| | *13.59* | *14.22* | *9.45* | *11.10* | *12.09* |
| San Antonio airport | 4.48 | 1.02 | 0.41 | 14.91 | 7.11 |
| | *10.18* | *8.58* | *5.92* | *6.12* | *9.32* |
| Victoria | 7.77 | 8.92 | 0.94 | 43.03 | 7.92 |
| | *12.85* | *11.10* | *8.25* | *7.82* | *12.52* |

Precipitation rates for a number of stations in central Texas are shown in Table 1. With the exception of the August data, all stations reported lower than average rainfall during these months apart from Houston Intercontinental Airport in June and July, and Austin International Airport in September (respectively north and northwest of Galveston Bay). Despite this, low salinities were found in June at the surface inshore and pushing southwards (Fig. 4a), with a strong, sloping salinity front between the surface layer and the deeper water. Salinity values across the front changed by ~12 psu along stations 18-20 and 21-23 just south of Galveston Bay. The salinity gradient decreased towards the south, with an inshore-offshore change of only 4 psu south of 28°N. The lowest surface salinity (station 21) was <22 at this time, and was still <32 along the southernmost line except at the outermost station. Bottom water salinities (not shown) were higher because of density stratification, with salinities of >35 found in water deeper than about 20m at stations in the eastern half of the grid and 35 m on the southern lines. The low surface salinities were most likely caused by westward flow from the Mississippi-Atchafalaya river system (MARS), together with local outflow from Galveston Bay. The MARS peaked





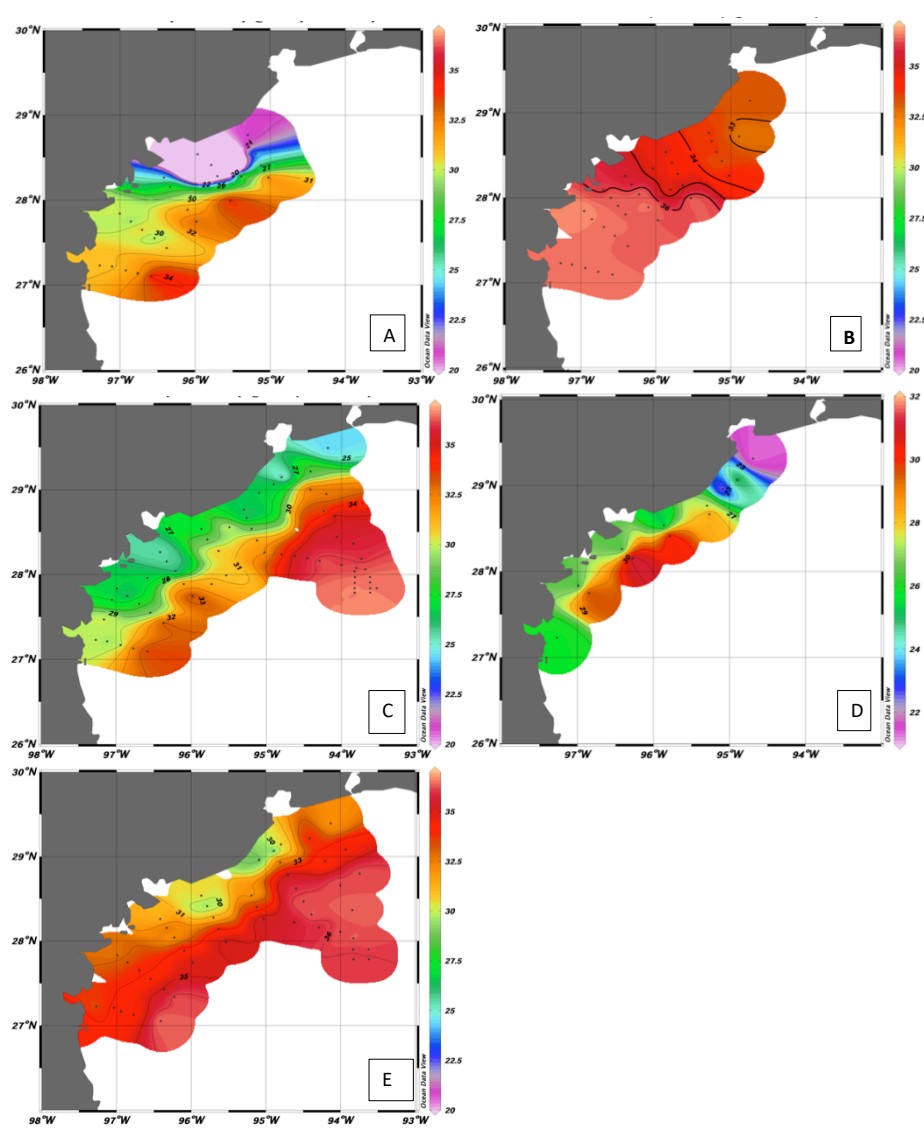

Fig. 4. Surface salinities during 2017 cruises in (a) June, (b) August, (c) September 22-27, (d) September 29 –
October 1, and (e) November.


during the 2017 spring flood at 1.22 Mcfs (34,500 m³/s), almost double the long-term mean from
1935-2017 (data from http://rivergages.mvr.usace.army.mil/, accessed 7.07.2021).




By August (Fig. 4b), surface salinities had increased across the region as a result of the southerly
winds, with a minimum of 32.15 just south of Galveston Bay, while the 35 surface isohaline was
situated off Matagorda Bay between stations 16-20 and stations 11-15. Bottom water was still
stratified at stations on the two northern lines, with salinities <35 only found at stations 16, 17,
21, and 22 and at the Wind Farm (29.14° N, 94.75° W). Further south, stations 1-10 and 13-15
all contained almost isohaline water with S > 36.

The fresh water from the hurricane caused a major change in the surface salinity by the time of
the first September cruise (22-27), resulting once again in a strong cross-shelf gradient (Fig. 4c).
Surface salinities were <33 throughout the region, apart from two stations at the extreme south of
the grid, where it was just above 33, and in the area more than 100 km offshore between
Galveston Bay and the Flower Gardens Banks, where there was a strong salinity front. A similar
situation was found a week later at the inshore stations (Fig. 4d), although the surface layer of
low salinity water had thinned and was confined to the innermost stations on each line. Vertical
sections in September showed very strong stratification of up to 10 psu within a 10-m depth
interval along all lines (e.g., Fig. 5; this section across stations 11-15, adjacent to Matagorda
Bay, is taken as representative for all five lines). The halocline was not flat, but deepened
towards the coast, giving a wedge of lower salinity water onshore, and the depth at which it
intersected the bottom decreased from ~30m in the north to less than 20m in the south. Water
with salinity > 36 was found at the bottom on all lines. By November (Figs 4e, 5), however, a
more typical salinity field was found, with well-mixed water throughout the coastal zone and a
general onshore-offshore gradient at all depths. This is normal for the region in the fall, when
atmospheric frontal systems tend to move across the Texas shelf and break down the summer
pycnocline (Cochrane and Kelly, 1986; Nowlin et al., 1998).

**3.4 Oxygen concentrations**
Oxygen concentrations in this region of the Gulf of Mexico are typically saturated above the
pycnocline, and this was the case during all four cruises, with concentrations between  210-220
$\mu$mol/L in June (not shown), when the surface temperature was around 25° C, and 190-215
$\mu$mol/L during August and September, when it was nearer 30° C (Fig. 5). By November, with







Fig. 5. Salinity (psu) and oxygen (μmol/L) sections across line 3 (stations 11-15) for the August (a), September (b) and November (c) cruises.





declining surface temperatures, the saturation concentration increased to between 210-230
μmol/L. Below the pycnocline, oxygen concentrations declined in the higher salinity water. This
effect was most pronounced offshore in June and August, when subtropical underwater, with





typical oxygen concentrations of 160-170 $\mu$mol/L, intruded onto the outer shelf (Fig. 5). Isolated
patches with concentrations <150 $\mu$mol/L were seen over the mid-shelf and across the eastern
part of the grid at this time. This situation had changed considerably by the time of the
September cruises, with bottom concentrations of 150 $\mu$mol/L or less over large parts of the
inner and middle shelf and at the outermost stations of the grid. Vertical sections showed that the
lowest oxygen concentrations occurred at the base of the pycnocline where it intersected the
seafloor (Fig. 5), but hypoxia (oxygen concentrations <62 $\mu$mol/L) was not observed at any
station. There was little change in either the pattern of oxygen distribution or concentrations at
the innermost stations between the two cruises in September (not shown). By November,
however, after the passage of a number of frontal systems with wind speeds up to 14 m/s, the
oxygen concentrations showed little vertical structure and the system could be said to have
returned to normal for that month.

**3.5 Nutrients**
Nutrient concentrations in the coastal waters along the Texas coast in summer are typically very
low at the surface, increasing with depth even on the shallow shelf as nutrient regeneration takes
place near the bottom. This is especially the case when hypoxic events occur (Nowlin et al.,
1998; DiMarco and Zimmerle, 2017; Bianchi et al., 2010). Mean concentrations in the upper
30m of the water column for all nutrients at stations within the grid as well as at additional
stations having water depths shallower than 50m are given in Table 2. Data from the second
September cruise, which covered only the two inshore stations on each line, are not included in
the table. These data showed similar patterns to the cruise a week earlier, although mean
concentrations were higher because of the proximity of the coast and the many freshwater
discharges from bays and rivers.

In higher salinity (>35) water and offshore, nutrient concentrations increase only slowly with
depth and nitrate and silicate concentrations > 5 $\mu$mol/L are generally found in midwater only
below depths of about 50 and 100m respectively (Fig. 6, Supplemental Fig. S3). Only one nitrate
sample (in September) containing more than 8 $\mu$mol/L came from below 60m depth. Nitrite
concentrations were almost all low, with mean concentrations in the upper 30m below 0.5
$\mu$mol/L on all four cruises, although individual surface concentrations were considerably higher.



Table 2. Mean, range and number of samples (N) for nitrate, nitrite, ammonia, phosphate and silicate in the upper
30m of the water column for all four cruises. DIN is calculated as the sum of the three nitrogen species. DIN:P and
DIN:Si ratios use the values for all individual samples.

|           |       | June | August | September | November |
|-----------|-------|------|--------|-----------|----------|
| Nitrate | Mean | 0.71 | 0.10 | 0.57 | 0.52 |
|           | Range | 0.00-10.60 | 0.00-1.98 | 0.00-7.41 | 0.00-1.98 |
|           | N | 85 | 94 | 194 | 164 |
| Nitrite | Mean | 0.43 | 0.18 | 0.44 | 0.36 |
|           | Range | 0.00-2.80 | 0.00-1.04 | 0.03-4.76 | 0.00-1.13 |
|           | N | 86 | 98 | 196 | 172 |
| Phosphate | Mean | 1.07 | 0.65 | 1.30 | 1.00 |
|           | Range | 0.21-2.85 | 0.00-3.55 | 0.00-5.63 | 0.00-3.24 |
|           | N | 85 | 91 | 190 | 169 |
| Silicate | Mean | 6.00 | 5.04 | 7.00 | 7.76 |
|           | Range | 1.18-26.89 | 0.00-20.09 | 0.00-40.23 | 0.94-25.71 |
|           | N | 84 | 89 | 193 | 168 |
| Ammonia | Mean | 1.90 | 3.74 | 2.39 | 2.91 |
|           | Range | 0.00-7.62 | 1.37-8.05 | 0.08-4.97 | 0.89-4.80 |
|           | N | 84 | 87 | 192 | 162 |
| DIN | Mean | 3.01 | 3.70 | 3.37 | 3.72 |
|           | Range | 0.01-14.47 | 0.14-8.56 | 1.02-12.35 | 1.05-7.03 |
|           | N | 85 | 95 | 191 | 160 |
| DIN:P |  | 3.56 | 11.95 | 4.98 | 10.11 |
|           | Range | 0.03-25.86 | 0.00-324 | 0.00-138 | 0.00-381 |
| DIN:Si |  | 0.63 | 2.59 | 1.17 | 0.78 |
|           | Range | 0.00-3.20 | 0.00-53.29 | 0.00-25.21 | 0.10-4.78 |

Ammonia concentrations were also variable, particularly inshore, and generally provided a
background concentration of about 2-4 $\mu$mol/L. As a result, DIN distribution closely resembled
that for nitrate but with the contribution from ammonia (Fig. S4).



A

B

C


Fig. 6. Nitrate and silicate ($\mu$mol/L) sections along line 3 (stations 11-15) during August (a), September (b) and
November (c) cruises.

Phosphate concentrations (not shown) were similarly lower at the surface than at depth, except in
September, when the influence of surface runoff gave rise to concentrations above $3\ \mu$mol/L in
the upper 10m of the water column and a background concentration between $1.5 - 3\ \mu$mol/L in





the rest of the water column up to 50 km offshore (between stations 13 and 14). Phosphate is
almost always non-limiting for phytoplankton in this region, so that residual phosphate
concentrations can be found even though nitrate is depleted (Bianchi et al., 2010), although
Sylvan et al. (2006, 2007) and Quigg et al. (2011) have suggested phosphate limitation can occur
further east in the Mississippi plume. Silicate, however, showed an opposite trend to the general
pattern of the other elements, with almost all samples >15 $\mu$mol/L coming from the upper 25m of
the water column, and concentrations decreased with depth to <5 $\mu$mol/L below 100m (Figs 6,
S3). Silicate also showed a cross-shelf gradient, particularly along the two southernmost lines
(not shown).

This general distribution shown in Figs. 5 and 6 was seen during early summer along all the lines
occupied during June and August.  In June, high concentrations of both nitrate and silicate were
seen at stations 21 and 22, immediately south of Galveston Bay, where bottom water oxygen
concentrations were < 90 $\mu$M/L; elsewhere midwater levels of both elements were low, with very
low nitrate concentrations (<0.5 $\mu$mol/L) being found even at the bottom at some stations. While
silicate concentrations were more variable, highest concentrations were typically again seen at
the bottom, and midwater concentrations were generally < 5 $\mu$mol/L. The situation was similar in
August (Fig. 6), when nitrate was very low throughout the region, and even bottom nitrate values
were below detection at many stations.

In September, despite the extreme freshwater runoff, nitrate concentrations were still low except
near the bottom at shallow stations, and there was little sign of any surface or mid-water increase
in concentration (Fig. 6). A comparison of nitrate concentration with depth gave essentially the
same distribution as during earlier cruises, although there were more samples within a depth
range of 10-30m showing concentrations above 2 $\mu$mol/L (Fig. S3). These samples came from
bottom samples at shallow stations where the oxygen concentration was reduced. The cross-shelf
gradient seen in silicate concentrations was more pronounced on this cruise, and concentrations
exceeded 10 $\mu$mol/L throughout the water column at all the inshore stations. However, by
November, concentrations of both nutrients had decreased considerably, although the offshore
silicate gradient was still present and concentrations > 10 $\mu$mol/L were found inshore (Fig. 6).

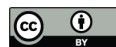



Phosphate concentrations higher than 2 $\mu$mol/L were seen only in September (Table 2),
suggesting the presence of terrestrial runoff following the hurricane.

Oxygen/nitrate and oxygen/silicate covariance plots are shown in Supplemental Fig. S5. High
nitrate values at oxygen concentrations greater than 200 $\mu$mol/L in August and September (22-
27) are from samples taken in low salinity surface water; where oxygen concentrations were
below 150 $\mu$mol/L the increase in nitrate concentration is caused either by regeneration over the
shelf or by the intrusion of deeper Subtropical Underwater. During these two cruises, higher
nitrate and silicate concentrations were associated generally with lower oxygen concentrations
(Fig. S5), although some surface samples on both cruises showed relatively high values,
associated with salinities < 35.

Quigg et al. (2011) state that DIN concentrations <1 $\mu$mol/L and a DIN:P ratio <10 indicate
nitrogen limitation,  with P <0.2 $\mu$mol/L and DIN:P >30 indicating P limitation and Si <2
$\mu$mol/L, DIN:Si >1 and Si:P <3 showing Si limitation. As shown in Table 2, DIN:P and DIN:Si
ratios for individual samples in the upper 30m of the water column were low during all four
cruises, with mean DIN:P being less than the 16:1 Redfield ratio throughout, while the mean
DIN:Si ratio was >1 only in the August and September cruises. This suggests both nitrogen
limitation throughout the period and possible silicate limitation of diatom growth during August
and September despite the background levels of ammonia that contributed to the DIN
concentration. While individual samples had higher ratios, these all occurred when either
phosphate or silicate concentrations were measurable but very low in comparison with DIN
concentrations (<0.1 $\mu$mol/L for P and <0.5 $\mu$mol/L for Si). The ratios of the mean
concentrations of DIN across the region to the mean concentrations of P and Si (e.g., 3.01:1.07
for DIN:P in June), were 2.81 and 0.50, 5.69 and 0.73, 2.59 and 0.48, and 3.72 and 0.48 for the
June, August, September and November cruises respectively, again suggesting nitrogen
limitation.

**4   Discussion**
One would normally expect that the amount of rainfall seen during Hurricane Harvey would
result in an exceptional flushing of nutrients into the coastal bays and the offshore coastal zone,





as found, for example, in Biscayne Bay, Florida, following Hurricane Katrina in 2005 (Zhang et
al., 2009), or in the Caribbean in 1998 following Hurricane Georges (Gilbes et al., 2001), leading
to short-lived phytoplankton blooms, but the data show very little sign of such a buildup
offshore, other than excess phosphate seen during the first September cruise (see above). The
coastal bays turned into freshwater lakes – Galveston Bay was flushed by about three to five
times its volume of freshwater (Du et al., 2019; Thyng et al. 2020) and this resulted in higher
concentrations of nutrients, particularly nitrate and silicate, as well as blooms of phytoplankton
and cyanobacteria within the bay (Liu et al., 2019; Steichen et al., 2020). DIN concentrations, in
particular, were greatly reduced two weeks after the hurricane had passed through the region and
were back to normal conditions by November (Steichen et al., 2020, Fig. 7; J. Fitzsimmons, pers.
comm.), with concentrations above 5 $\mu$mol/L only found in the uppermost parts of the system
after about 15 September. Silicate concentrations similarly dropped quickly within the first two
weeks, although they remained above 40 $\mu$mol/L throughout the bay during the sampling period.

Following hurricane Harvey, low-oxygen water containing <160 $\mu$mol/L and nitrate
concentrations of > 2 $\mu$mol/L penetrated further onto the shelf during September than during
either August or November (Figs. 5, S3). The high salinity of this water mass (>36, Fig. 5)
suggests that it was Subtropical Underwater, which is found above 250 m in the northern Gulf
with typical core salinity of about 36.4 -36.5 near 100m depth in this region, and oxygen and
nitrate concentrations of about 110-150 $\mu$mol/L and 6-15 $\mu$mol/L respectively (Nowlin et al.,
1998). However, given the strong pycnocline shown by the salinity section (Fig. 5), there was
little opportunity for these additional nutrients to reach the surface layer and affect
phytoplankton production, and there is no evidence that such upwelling has resulted in hypoxia
in the past in this region.

Further south, the Matagorda-San Antonio-Aransas-Corpus Christi Bay system similarly showed
rapid short-term nutrient increases followed by hypoxia (Montagna et al., 2017; Walker et al.,
2021), but these were back to pre-storm concentrations by early October (Walker et al., 2021).
The levels in Guadeloupe Bay, an offshoot of San Antonio Bay, were followed at fortnightly
intervals from mid-August to mid-October and showed a rapid increase in nitrate but slower
increases in phosphate and silicate. This is not unexpected, given the solubility of nitrate ions



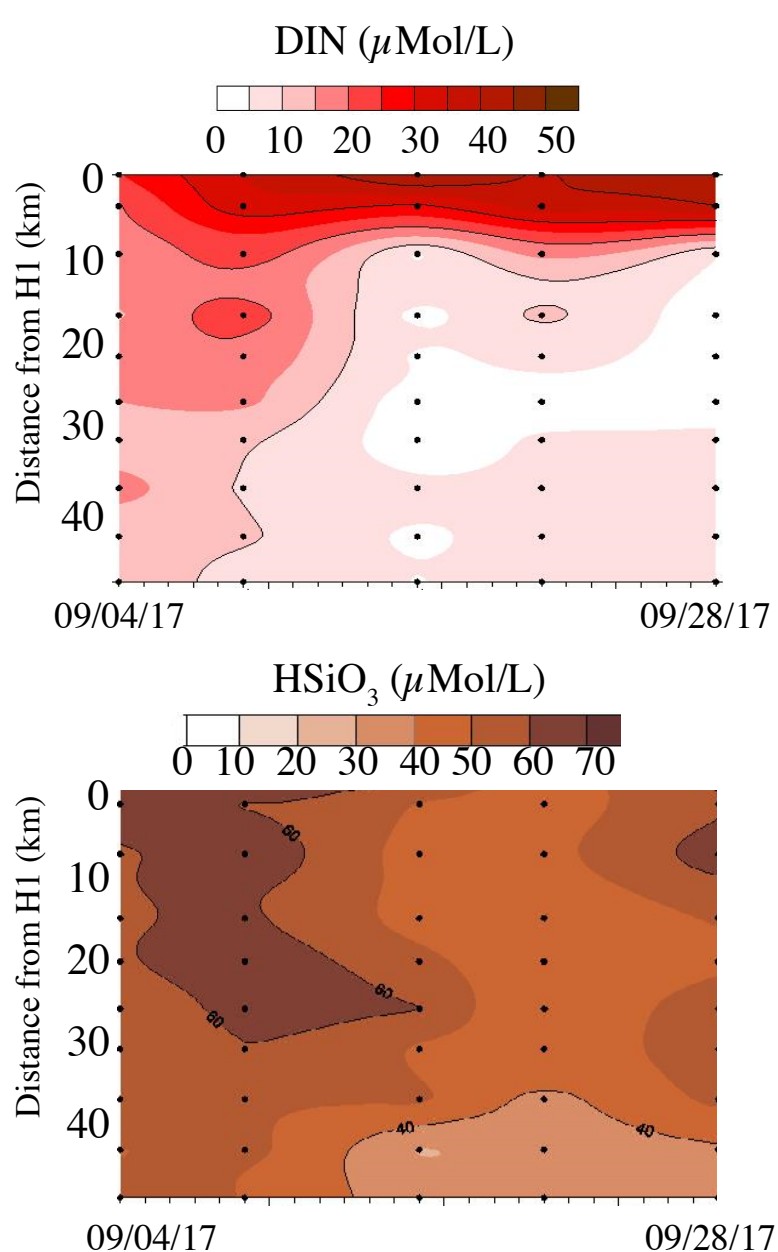

Fig. 7. Nitrate plus nitrite (a) and silicate (b) concentrations (μmol/L) measured along a transect through Galveston

Bay. Sampling dates were 9.04.17, 9.09.17, 9.16.17, 9.21.17, and 9.28.17. Station H1 was the innermost station in

the bay (see Steichen et al., 2020 for details).

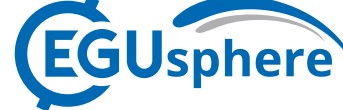


relative to the other two. Thus, it appears that the increases in nutrient concentrations affected
mainly the coastal bays and estuaries rather than the offshore coastal zone. This backs up
conclusions of Sahl et al. (1993) following a cruise along the Louisiana-Texas shelf in March
1989 when river discharges were at their highest levels during that year. They found that
nutrients derived from bay systems dissipated within about 20 km of the bay mouths, and that
higher nutrient concentrations below 80 m depth resulted from upwelling along the shelf edge, in
agreement with the work of Chen et al. (2003) and Walker et al. (2005). Similar retention of
nutrients, leading to short-lived (2-3 weeks) phytoplankton blooms, has been reported following
hurricanes in the Neuse River/Pamlico Sound system in N. Carolina (Paerl et al., 2001, 2018;
Peierls et al., 2003), in Chesapeake Bay (Roman et al., 2005), and Biscayne Bay (Zhang et al.,

2009).


Although nutrient fluxes were undoubtedly greatly increased immediately following the
hurricane, nutrient concentrations in Texas rivers are only sampled infrequently, and data do not
exist to allow us to calculate the overall fluxes during this period. However, the available data
suggest that absolute concentrations did not change very much following the hurricane in most
instances (Table 3). The infrequent sampling by federal and state authorities, coupled with the
rapid decrease in river flow by about September 7 (Fig. S1), suggest that nutrient concentrations
in the coastal bays and the coastal ocean were likely diluted by the time of our survey in late
September. Du et al. (2019) point out that while the salinity at the mouth of Galveston Bay was
back to normal about two weeks after the storm, it took almost two months to recover at stations
further inside the bay and the same time period at offshore buoys. Similar effects are likely at
other bay sites along the Texas coast.

Changes in salinity during and following the storms were recorded at offshore moorings. During
the period around the passage of the hurricane, the TABS moorings showed rapid decreases in
salinity with a slow increase thereafter (data not shown). Buoy X (offshore) showed the least
variability, with salinities remaining near 36.4 until 9.04.17, dropping briefly to 35.3, but
recovering to above 36 again by 9.06.17. Buoy D, inshore near Corpus Christi, also recorded





Table 3. Nutrient concentrations in Texas rivers around the time of the hurricane ($\mu$M/L). Data taken from USGS
and the Texas Commission on Environmental Quality (TCEQ) Clean Rivers Program for individual river basins.

a.   Trinity River (Baytown; USGS site 08067525)

| Date | Nitrate | Phosphate | Silicate |
|---|---|---|---|
| 7.06.17 | 10.15 | 2.03 | 74.2 |
| 7.19.17 | 11.28 | 2.52 | 90.0 |
| 8.15.17 | 11.43 | 3.16 | 155.5 |
| 9.05.19 | 10.64 | 1.74 | 96.0 |
| 11.08.17 | 5.43 | 1.58 | 143.5 |


b.   Trinity River (Liberty, USGS site 08067000)

| | | | |
|---|---|---|---|
| 8.16.17 | <2.86 | 2.38 | 137.5 |
| 8.31.16 | 8.71 | 1.32 | 97.8 |
| 9.05.16 | 15.85 | 2.26 | 127.0 |


c.   Brazos River (US 290; TCEQ site 11850)

| | | |
|---|---|---|
| 7.26.17 | 41.40 | <1.29 |
| 8.22.17 | 7.86 | <1.29 |
| 9.27.17 | 12.86 | 2.26 |
| 10.25.17 | 37.86 | 2.90 |


d.   Colorado River (La Grange; TCEQ site 12292)

| | | |
|---|---|---|
| 6.06.17 | 2.86 | 92.58 |
| 8.08.17 | 2.86 | 118.06 |
| 10.02.17 | 2.14 | 86.45 |


e.   San Antonio River (Goliad; TCEQ site 12791)

| | |
|---|---|
| 7.19.17 | <3.57 |
| 9.06.17 | <3.57 |
| 11.01.17 | <3.57 |


salinities of about 36.6 until 8.23.17, dropping to 34.7 on 8.26, but were >36 a day later.
Salinities dropped again on 8.29, remaining in the range 32-34 until 9.06, after which they
dropped again to below 30, where they remained until 10.24.17, with a minimum salinity of
20.51 on 9.13. Further up the coast buoys B and F both experienced decreased salinities (buoy W
did not record salinities during the passage of the hurricane). Before the hurricane, salinities in
this region were in the range 32.5-34.5, with the higher salinities offshore. Following the passage
of the storm, buoy F recorded a minimum salinity of 15.25 on 9.01.17 and salinities <20 until
9.06.17. A salinity of 30 was only recorded again here on 9.08.17. The inshore buoy B recorded
minimum salinities in the range 19-21 on 8.30. These remained <23 until 9.09, and below 30 for



527 the remainder of the month, after which they increased again to around 32. The fact that the

528 minimum salinity was recorded at the offshore mooring is presumably related to the strength of

529 the plume emanating from Galveston Bay with enough momentum to overcome the Coriolis

530 force that would tend to push it to the southwest close to the coast (Du et al., 2019).

532 These data suggest that there was a slow southward movement of low salinity water along the

533 coast (see Figs. 4c, d) after the hurricane as the coastal current was re-established. The fact that

534 winds were from the east for almost the whole of September would have assisted this downcoast

535 movement, as described by Cochrane and Kelly (1986). Mixing during the infrequent northerly

536 wind bursts would cause salinities to increase again, although even in November salinities below

537 30 were still seen between Galveston Bay and Matagorda-Corpus Christi Bays (Fig. 4e).

539 The effects of the hurricane on phytoplankton productivity, as measured by chlorophyll

540 concentrations along the Texas shelf and slope, were examined using both in situ fluorescence

541 data obtained during the cruises and satellite imagery from the MODIS sensor on the Aqua

542 satellite (Fig. 8). The Texas coast and northwestern Gulf of Mexico were covered with clouds

543 during the pre-Harvey and post-Harvey cruises, however a time-history of four high quality

544 chlor-a images on August 18 (pre-Harvey), September 2 (6 days post-Harvey), September 11 and

545 September 16, 2017 (Fig. 7) revealed shelf events between the two cruises closest to Harvey's

546 landfall.

548 During mid-August, the highest chlorophyll-*a* concentrations and the maximum offshore extent

549 of blooms were found off central Louisiana, seaward and southwest of the Atchafalaya Bay

550 system, but within the 20m isobath. The zone of pigmented water extended to the 20 m isobath

551 southwest of Atchafalaya Bay but narrowed significantly from Sabine Lake (93.83°W) to Port

552 Aransas Bay (97°W). This distribution likely resulted from the pre-storm input of nutrients from

553 the Atchafalaya and Mississippi Rivers onto the shelf coupled with generally low summer flows

554 from Texas rivers. In contrast, by 2 September the highest chlorophyll-*a* concentrations were

555 detectable along the Texas coast between Sabine Lake and extending southwestwards to at least

556 Corpus Christi Bay. The widest zone of pigmented water extended well beyond the 20 m isobath

557 east, southeast, and south of Galveston Bay, likely due to the flux of nutrients and terrestrial



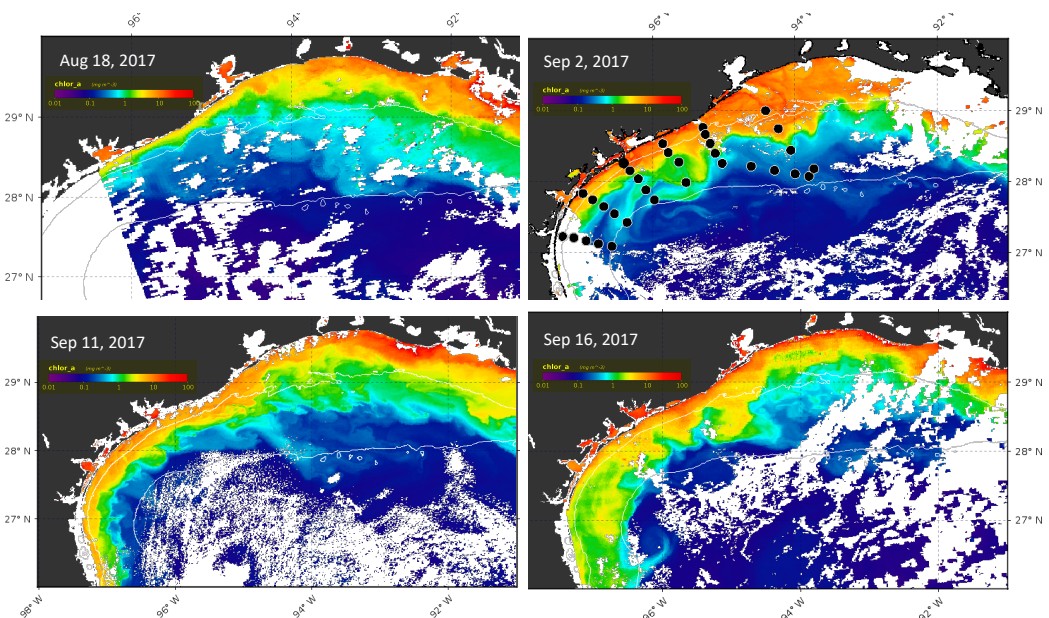

Fig. 8. Aqua-1 MODIS imagery depicting chlorophyll *a* estimates for August 18, September 2, September 11 and September 16, 2017. White areas along the Louisiana shelf and offshore are clouds. Thin white lines denote 20m and 100m isobaths. Station positions are indicated by the black dots on the 2 September image.

colored material from coastal areas flushed onto the shelf (Du et al., 2019). Maximum satellite-derived coastal chlorophyll-*a* values near Galveston Bay were 16 mg m$^{-3}$. Moving seaward to the 20 m isobath, values decreased to 10 mg m$^{-3}$ and fell below 1 mg m$^{-3}$ on the 100 m isobath (Fig. 8). The accumulation of pigmented water with high chlorophyll-*a* concentrations between Galveston Bay and Calcasieu Lake (93.45°W) likely resulted from convergence of the downcoast Louisiana river waters (Quigg et al., 2011) with upcoast hurricane-related discharges from the Galveston Bay region, as surface currents at TABS buoy B were offshore and decreased from ~75 cm/s to 20 cm/s during the period from 30 August to 3 September (Fig. 3).

By September 11, the zone of pigmented water on the shelf near Galveston had retreated shoreward and chlorophyll-*a* concentrations had decreased somewhat along the Texas coast, while the highest concentrations were observed further east along the Louisiana coast near Atchafalaya Bay (Fig. 8). Chlorophyll-a concentrations increased along the Texas coast by 16 September, accompanied by offshore movement in several lobes that reached the 100 m isobath





between Corpus Christi and the Mexican border at 26° N, but highest concentrations were only
about one tenth of those seen immediately after the storm. During the latter half of the month, the
maximum concentration of chlorophyll continued to decrease slowly and likely moved further
south and offshore under the influence of the prevailing currents (Fig. 3). Our first post-storm
cruise occurred between 22-27 September, so we would have missed the maximum extent of the
bloom and presumably also missed any offshore nutrient maximum that fueled the phytoplankton
growth. There was no evidence for upwelled nutrients resulting in blooms at the shelf edge, as
reported off Louisiana following Hurricane Ivan in 2004 (Walker et al., 2005) or in the East
China Sea by Chen et al. (2003).

In contrast to the satellite-derived data, fluorescence data from the CTD casts taken during all
cruises were much lower, especially in the upper mixed layer, where concentrations were
invariably <1 $\mu$g/L. During the period of the 22-27 September cruise measurements at only 4 of
37 stations had concentrations >1.0 mg m$^{-3}$, while at 29 stations they were 0.5 mg m$^{-3}$ or less.
The maximum concentration of 1.7 mg m$^{-3}$ was found inshore just south of Galveston Bay.
Midwater maxima only exceeded 2 mg m$^{-3}$ at offshore stations 27 and 28, where the maxima
were found at depths below 40m. This is similar to summer conditions reported by Nowlin et al.
(1998) and to previous data we have collected during summer cruises in the northern GoM.
Three days later, however, when the inshore stations were reoccupied, mean fluorescence values
showed 1-2 mg m$^{-3}$ at all inshore stations, with concentrations up to 4.8 mg m$^{-3}$ immediately
offshore of Galveston in the plume. The steady wind conditions during the latter half of
September (2-6 m/s from the SE) may account for the difference between the two cruises, as
there was no mixing of the surface water and so only limited phytoplankton growth could take
place. As stated above, the discrepancy between satellite-derived and in situ values may be
related to CDOM interference in the satellite estimates, with the higher concentrations in early
September shown in Fig. 8 resulting at least partly from the hurricane stirring up bottom
sediments in the shallow coastal zone.

Although September is normally the month when seasonal hypoxia (oxygen concentrations <62
$\mu$mol/L) in the northern Gulf of Mexico ends, because of the passage of storm fronts, the strong
stratification resulting from the freshwater input might have been  expected to reduce oxygen



concentrations below the pycnocline. Hypoxia in the northern Gulf of Mexico is generally
assumed to have three requirements: a high supply of nutrients, especially nitrogen, from rivers
or other terrestrial runoff, stable stratification with a mid-water pycnocline, and relatively low
wind conditions (Bianchi et al., 2010; Rabalais et al., 2007; Wiseman et al., 1997). This
combination of factors allows large phytoplankton populations to develop in the euphotic zone,
resulting in subpycnocline oxygen consumption during microbial respiration as the
phytoplankton die and sink. The presence of the pycnocline inhibits oxygen diffusion into the
bottom layer, resulting in oxygen depletion and eventually hypoxia, and relatively high
concentrations of organic matter can build up in the sediments, possibly leading to a reservoir of
labile material that contributes to hypoxia the following year during resuspension of the bottom
boundary layer (Hetland and DiMarco, 2008; Turner et al., 2008). Rabalais et al. (1999) state that
hypoxia can in fact occur in almost any month if conditions, particularly stratification, are right.
While the most intense hypoxia occurs over the Louisiana shelf (Rabalais et al., 1999), dissolved
oxygen levels below 30 $\mu$M/L have been detected during NOAA SEAMAP cruises as far west as
96°W, with occasional samples between 30-60 $\mu$M/L identified near Corpus Christi (see
https://www.ncei.noaa.gov/maps/gulf-data-atlas/atlas.htm), as well as following local flood
events (DiMarco et al., 2012; Kealoha et al., 2020), and bacteria from terrestrial sources have
been found in sponges at the Flower Gardens Banks National Marine Sanctuary near 28°N,
29.5°W (Shore et al., 2021).

Although hypoxia off the Texas coast has typically been linked to southwestward advection from
the Mississippi and Atchafalaya Rivers, high flow rates from local rivers have also been
implicated (Harper et al., 1981; Pokryfki and Randall, 1987; DiMarco et al., 2012). Di Marco et
al. (2012), using oxygen isotope data, specifically related hypoxia off the Brazos River mouth in
July 2007 to strong stratification following river discharge rates an order of magnitude higher
than normal. During the passage of Hurricane Harvey, the water column was completely mixed
and saturated in oxygen, but the torrential rainfall led to runoff that created a stable pycnocline,
and calm conditions after the storm meant that phytoplankton growth was possible. Further east,
on the Louisiana shelf, stratification is re-established within a few days of the passage of storm
fronts or hurricanes and bottom water oxygen depletion can begin rapidly once the storm has
passed (e.g., Bianchi et al., 2010; Jarvis et al., 2021). However, despite the strong stratification

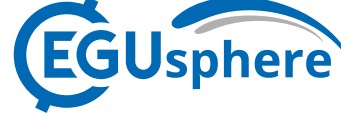

observed along the Texas coast, the two particularly interesting points found following our
cruises were the lack of any obvious signs of hypoxia over the Texas shelf, and the apparent lack
of increased nutrient concentrations, other than phosphate, in coastal water following the passage
of the hurricane. Plotting the difference in salinity between surface and bottom samples, a
measure of water column stability (DiMarco et al., 2012), against bottom oxygen concentrations
during the September cruise gave only a low correlation, with $R^2 = 0.15$ (n = 38), as opposed to
the 0.79 (n = 14) reported in 2007 by DiMarco et al. (2012). This suggests that stratification by
itself was not responsible for the observed bottom oxygen concentrations over the shelf.

While hypoxia occurrence largely follows local nutrient supply, low oxygen waters can also be
injected onto the shelf from offshore through upwelling. Chen et al. (2003), for example,
suggested that hypoxia in the East China Sea could be induced by the "cross-shelf upwelling of
nutrient-rich Kuroshio water after the passage of typhoon Herb in a normally downwelling
region." This led to increased primary productivity following the passage of the typhoon. While
they agreed with Shiah et al. (2000) that terrestrial runoff was a factor in increased local coastal
productivity following the storms, they suggested that the upwelling of subsurface Kuroshio
water was equally important. The increased upwelling was thought to result from "a larger
buoyancy effect caused by the rains as well as the shoreward movement of the Kuroshio caused
by the typhoons."

The lack of hypoxia following Hurricane Harvey can therefore perhaps be explained by four
factors. First, only a limited flux of nutrients made it out of the bays and into the coastal zone,
where it was likely taken up rapidly by phytoplankton, as seen elsewhere. Additionally,
southward and offshore advection of low salinity runoff increased the rate of dilution through
mixing with pre-existing low-nutrient surface shelf water. The largest bay systems have
relatively narrow entrances, which reduce the rate at which the fresh water can escape – the main
entrance to Galveston Bay, which includes the deep, dredged Houston Ship Channel, is only 2.3
km wide and the turnover time for water is 15-60 days under normal conditions, with shorter
periods coinciding with flood conditions (Solis and Powell, 1999; Rayson et al., 2016). Thyng et
al. (2020) have estimated that the initial flushing of Galveston Bay during Hurricane Harvey
took only 2-3 days following the initial heavy rainfall. For the Corpus Christi Bay/Aransas Bay





670 system the turnover time under normal conditions is estimated to be more than 300 days (Solis

671 and Powell, 1999), similar to Pamlico Sound (Paerl et al., 2001).


673 Second, the sheer volume of water rapidly removed available soluble nutrients within the first

674 few hours so that runoff later during the storm was essentially pure rainwater. It is known that

675 large percentages of available nutrients are removed in stormwater runoff in the first minutes or

676 hours following a downpour and concentrations then drop (e.g., Cordery, 1977; Horner et al.,

677 1994; Fellman et al., 2008). Similar effects have been reported for trace metals in the floodplain

678 of the Pearl River in Mississippi (Shim et al., 2017), where maximum downstream

679 concentrations were not found following peak flows. These authors suggested that the rapid

680 flushing overwhelmed the rate at which soluble metal-organic complexes could be regenerated.

681 As the hurricane occurred in late summer, any nutrients applied to cropland along the Texas

682 coastline in spring would largely have been taken up by the vegetation and so be unavailable for

683 washout. While Corpus Christi (population ~325,000) and Houston (~4 million) are large

684 population centers with multiple sewage treatment plants that flooded following the hurricane,

685 both are sited upstream of large bay systems that would have attenuated the speed at which

686 stormwater runoff dissipated. The rate of change of nutrient concentrations in Galveston Bay

687 (Fig. 7) shows that uptake within the bay system was likely considerably more important than

688 flushing, even with the apparently short flushing time calculated by Thyng et al (2020).

690 While nutrient flushing was reduced following the hurricane, the same is unlikely to be true for

691 sediment. As shown in Fig. S2, and as discussed by D'Sa et al. (2018), Du et al. (2019), and

692 Steichen et al. (2020), large sediment plumes occurred off the mouths of major bays and rivers.

693 The heavy sediment loads would have not only increased the turbidity of the water column and

694 thereby reduced light intensity in the euphotic zone, but could also have led to reduced phosphate

695 concentrations as phosphate is known to bind to sediment particles (e.g., Suess, 1981). Both

696 factors would have contributed to reduced phytoplankton production, which is a major factor in

697 hypoxia formation (Bianchi et al., 2010). While phosphate concentrations in the coastal zone

698 were highest during the first September cruise, suggesting at least some terrestrial runoff

699 immediately following the hurricane and possibly desorption from suspended sediment, the low

700 nitrate concentrations seen during this cruise and the low chlorophyll fluorescence suggests only





a short-term phytoplankton bloom at most, again similar to previous observations (e.g., Roman et
al., 2005).

The final potential control is sediment composition along the Texas shelf. Most sediments in this
region are coarse, sandy, and contain little organic matter (Hedges and Parker, 1974). This is in
contrast to the Louisiana shelf, where muddy, organic sediments are quite common and act as a
reservoir of material that can continue to reduce oxygen concentrations once stratification is
established (Bianchi et al., 2010; Corbett et al., 2006; Eldridge and Morse, 2008; Turner et al.,
2008). This is especially true within coastal embayments, such as Terrebonne Bay, LA, where
the organic carbon content can exceed 5% thanks to organic matter input from the surrounding
marshes and swamps (Hedges and Parker, 1974; Bianchi et al., 2009, 2010). Even near the
Mississippi and Atchafalaya Rivers, however, typical organic carbon sediment content on the
shelf is generally <2% (Gordon and Goni, 2004; Gearing et al, 1977), while further west off the
Texas coast it is typically < 1% (Hedges and Parker, 1974, Bianchi et al., 1997). This suggests
that organic matter along the Texas shelf is refractory, and less likely to add to any oxygen
demand, and that hypoxia on the Texas shelf is generally driven by water column respiration as
discussed by Hetland and DiMarco (2008). In this region stratification alone is not sufficient to
bring about hypoxic conditions in the absence of high nutrient concentrations and phytoplankton
blooms.

**5   Conclusions**
Although Hurricane Harvey led to pronounced flooding and exceptional freshwater runoff along
the Texas coast, it did not lead to lasting high nutrient concentrations offshore, largely because of
dilution by the rainfall, and the likely rapid uptake by phytoplankton of nutrients washed out of
the bays. The most pronounced changes in nutrient concentrations were seen in the coastal bays.
Even here, changes from background levels were short-lived, and conditions were essentially
back to normal by November, some eight weeks after the hurricane, following northerly wind
bursts that caused mixing within the water column. While a transient bloom of phytoplankton
was observed in satellite imagery offshore following the hurricane, its short existence suggests
that hypoxia could not develop despite the stratification because nutrient concentrations were not
high enough to support continued phytoplankton productivity. Similarly, the lack of an organic





matter reservoir in the shelf sediments means there is no additional oxygen demand in Texas
bottom waters, and hypoxia here depends on water column decomposition.

**6 Acknowledgements**
We are grateful to the Captains and crews of the R.V. *Manta* and R.V. *Point Sur* for their
excellent service during the cruises, and to the enthusiasm of the students and technicians who
helped with data collection. The TABS system is funded by the Texas General Land Office and
operated by the TAMU Geochemical and Environmental Research Group. Cruises were funded
by the Texas Governor's Fund through the Texas OneGulf Center of Excellence and an NSF
RAPID award (OCE-1760381) to Drs. Knap, Chapman and DiMarco. A.H. K would also like to
acknowledge financial support from the G. Unger Vetlesen Foundation. We thank Ysabel Wang
for help with the figures, and Alaric Haag for assistance with SeaDAS image processing. Walker
and Haag thank the Gulf of Mexico Coastal Ocean Observing System (GCOOS) for funding
LSU Earth Scan Laboratory activities. Bathymetry shown in satellite imagery was provided by
GEBCO Compilation Group (2020) GEBCO 2020 Grid (doi:10.5285/a29c5465-b138-234d-
e053-6c86abc040b9). Funding sources had no involvement in study design, data collection and
interpretation, or manuscript preparation.

Data are being submitted to the Biological and Chemical Oceanography Data Management
Office (BCO-DMO). The title and DOI for the first set are: Processed CTD profile data from all
electronic sensors mounted on rosette from R/V Pt. Sur PS 18-09 Legs 01 and 03, Hurricane
Harvey RAPID Response cruise (western Gulf of Mexico) September-October 2017
(DOI:10.26008/1912/bco-dmo.809428.1).

**7 Credit author statement**
The project was conceptualized by SFD and AHK; PC and SFD conducted investigations on all
cruises and collected and analyzed the initial data; AQ provided data from Galveston Bay; NDW
provided satellite imagery. PC wrote the initial draft; all authors provided comments and edits.
The authors declare that they have no conflict of interest.



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
