# Peer review of "The effects of Hurricane Harvey on Texas coastal zone chemistry"

_EGUsphere, 2022_

## Author Response (AR1)

**Responses to reviewers**

*We thank the reviewers for their comments. Responses to individual comments are covered here as listed, in italics. In the revised text, we have used **bold** for new material, and strikeouts where we have deleted wording.*

**RC1**: 'Comment on egusphere-2022-1075', Anonymous Referee #1, 24 Oct 2022  reply
Chapman et al. provide results from hydrographic/nutrient/oxygen surveys of Texas coastal waters following Hurricane Harvey in 2017. Episodic events are often thought to have disproportionate environmental impacts, so this sort of study is welcome. The authors were fortunate to have done some coastal surveys before the storm. And, while one might have wished for a more detailed time series after the storm, the limits of obtaining funding, supplies, and shiptime meant that their first post-storm cruise occurred several weeks after the storm. Having been in a similar situation myself, I can appreciate their predicament. Furthermore, their results basically indicate that nothing 'big' happened offshore biogeochemically. That is, there didn't appear to be a big enrichment of nutrients nor any resulting hypoxia (though, if something big and brief happened before they could get out sampling, they would have missed it). Nonetheless, observing a muted response to a major stimulus is an important result. So, this is ultimately publishable and will be of interest to the community.

*Thank you for realizing how hard it is to get a proposal written and find a ship in short order after a situation like Harvey!*

My main quibble with this manuscript is that it is too lengthy (~9000 words) for what is shown. For instance, there's a presentation of satellite chlorophyll images with a couple pages of discussion and then field fluorescence data and then finally an acknowledgment that the discrepancy between satellite and inferred in situ chlorophyll likely reflects CDOM interference on the images. Likewise there's a fair bit of text about why no hypoxia was observed on the shelf following the storm....since this is all fairly speculative, I think it could and should be condensed. That said, one topic that could use a little commentary (i.e., just a few sentences) is comparisons with other post-hurricane coastal studies. Just a brief 'what's similar/what's different' summation could be of interest.

*We have cut the text where possible, and attempted to condense it as suggested by the reviewers.*

Some additional comments:

1. At various spots in the text, including tables, numbers are presented with two much

implied precision (significant figures). In some cases this may have resulted from a conversion from English units to metric (e.g., the <1.29 µM PO4 in Table 3 which was probably originally <40 µg-P/L).

*Actually, we make all our measurements in µmol/L, rather than µg/L, as in Table 2, so it seemed sensible to convert USGS and TCEQ numbers in Table 3 to µmol/L to be consistent. Whether we have given too much precision would seem to be a matter for debate. We normally use 2 significant figures.*

2. A number of the figures have text in too small a font to be able to read. Even ODV can have its default font settings changed (and, you can even get it to show a µ rather than a u). Also, for the figures that have multiple parts representing different cruises, it would be nice if the dates were shown on each sub-plot rather than making the reader go to the caption.

*We have done our best to improve the font size as suggested, and added dates to the figures themselves.*

3. A minor point, but in Table 1, the caption starts with "Precipitation rates (cm)". I am afraid that cm is an amount, not a rate. I suspect these numbers are cm/mo.

*You are correct. This has been changed.*

4. On Fig. 4, there are some contour wobbles near the coastline that are not supported by data. Adjusting the ODV contouring parameters or masking out the near shore might be helpful.

*We purposely used the same x and y scales on all the components of Fig. 4. We are not really sure which contours the reviewer refers to in this instance.*

5. Table 2: no units are provided.

*We have added the units, using µM as per previous comment.*

6. Lines 348-350: Ammonia was variable and in greater abundance than nitrate, but DIN follows nitrate? I am confused.

*Apologies for the confusion. What this means is that the overall distribution of total nitrogen is qualitatively similar to that for nitrate, but with the addition of background concentrations of 2-4 µmol/L in most of the water column. Ammonia made a larger contribution near the sediments where overall nutrient concentration were higher. We have rewritten these sentences to say:*

*Ammonia concentrations were variable, particularly inshore, **but** generally provided a background concentration of about 2-4 µmol/L.  As a result, DIN distribution resembled that for nitrate but with the **added background** contribution from ammonia (Fig. S4).*

7. There are lots of little grammar and style errors. For instance, µM/L is used in a few spots (including the text, a table, and a figure)...it should either be µM or µmol/L.
*The quantities have been corrected to show µmol/L throughout. We have gone through the manuscript again and hope we have corrected any other such errors.*

8. Lines 425-432: The authors seem to be saying that nutrients were removed in the bays fairly rapidly by blooms. But also, they indicate that Galveston Bay was flushed with 3-5x its volume of freshwater. And, later in the paper they allude to a flushing or rate-limitation effect in which the initial runoff is high in nutrients and then the source gets depleted. So, couldn't the fall in bay nutrients have resulted from the rapid flushing of the bay by a diminishing nutrient source?
*We don't believe so in this case. Texas bays are generally oligotrophic during the summer, because there is little riverine input, and the coastal waters also have very low nutrient concentrations at the surface (as shown by our pre-hurricane data). So the fresh water that replaced the brackish/salty water in the bay prior to the storm brought in more nutrients than were there previously. The papers by Liu et al (2019) and Steichen et al (2020) that we cite also state that phytoplankton blooms played a major part in reducing the nutrients in the bay after the storm passed. We have altered the wording to read: "Since Texas bays are oligotrophic during the summer, this influx of freshwater resulted in higher concentrations of nutrients...."*

9. Lines 445-446: This is a little confusing. Previously you've said that you didn't see shelf hypoxia. Now you mention that some other nearby systems "similarly showed rapid short-term nutrient increases followed by hypoxia". I think this just needs some slight rewording because "similarly" implies that you did see hypoxia in your study area.
*We have rewritten this sentence to read: "Further south, the Matagorda-San Antonio-Aransas-Corpus Christi Bay system also showed rapid short-term nutrient increases, followed in this case by hypoxia (Montagna et al., 2017; Walker et al., 2021), but nutrient concentrations were back to pre-storm concentrations by early October (Walker et al., 2021)."*

10. Figure 7: a location map would be helpful. Also, are these surface or bottom samples?
*We have added a map of station positions in Galveston Bay to Fig. 1. All samples were surface samples. Except for the ship channel, Galveston Bay is almost all less than 10m deep.*

11. Lines 450-457: "This is not unexpected given the solubility of nitrate ions relative to the other two." Please provide a reference.
*This has been rephrased to read "This is not unexpected, given that nitrate does not bind readily to sediment particles or organo-iron complexes like phosphate and silicate (Lewin, 1961; Suess, 1981).*

12. Line 544: chlorophyll a
*Corrected*

13. Hopefully, the note in the acknowledgments that the data 'are being submitted' will be updated to 'the data are available at' by the time of any revisions to the manuscript.

**Reviewer 2 General Comment:**

To understand the effects of Hurricane Harvey on Texas coastal zone chemistry, especially inorganic nutrients and dissolved oxygen. The authors used five cruises measurements, two pre- and three post-hurricane, with data on temperature, salinity, dissolved inorganic nutrients, DO, and fluorescence at stations along Texas coastal zone, Gulf of Mexico. In addition, both wind and current data were retrieved from website of the TABS moorings system along the Texas coast and the National Data Buoy Center. To evidence the phytoplankton dynamics at pre- and post-hurricane periods, satellite imagery was used and downloaded from the NASA Goddard ocean color website. I appreciate and admire the amount of data set of this ms.

I appreciate they provided a lot of data set, especially the field data. The results showed that even with intense terrestrial runoff due to rainfall caused by hurricane, nutrient supply to the coastal ocean was transient and with little phytoplankton growth and no hypoxia in coastal zone after hurricane. To demonstrate and explain those results and outcomes, a lot of data has been applied. However, I do find out that the whole manuscript mostly only stated the observed results but lacked of the significant scientific evidences, especially without supporting by statistical evidences. In addition, the presentation of this ms should be in a more logistic way and provided the organized and sorted information; the unnecessary details normally will distract the way of thinking and reading of the reader. In the text, there are also many local names has been addressed when described the data, and they should be clearly labeled in the map. Otherwise, it will confuse the international reader, especially for whom do not familiar with this region. Overall, I do feel this ms was very descriptive and it should be concise and focus on the main theme of this Hurricane effect with supporting evidences from this study.

*We thank the reviewer for these comments and have tried to improve the text by shortening it where possible. We are not sure what it meant by "The manuscript...lacked scientific evidences, especially without supporting by statistical evidences." We have used all the data*

*that we have, what additional statistics are needed? We have also added a number of local names to Fig. 2.*

Specific comments:

Introduction:

In the introduction, you might want to focus on what we have known about the phenomena, e.g., variables or ecosystems response to hurricane's impact. Therefore, you may consider to simplify or trim the unnecessary details regarding description on hurricanes and Hurricane Harvey.
*We have put a paragraph at the beginning of the Discussion section that covers how hurricanes are likely to affect the coastal zone. This collects together the various comments on this topic that were scattered throughout this section. We have also added a reference to a new paper on local acidification from the freshwater input (Hicks et al., 2022).*

lines 31-39. This can be simplified by one sentence.
*We have reduced this paragraph to " The Gulf of Mexico is renowned for its hurricanes and tropical storms, and 2017 was **a very** active year in the Atlantic, with 10 hurricanes and 8 tropical cyclones and depressions." This then runs on to the existing paragraph 2.*

lines 55-59, again, too much unnecessary details on Hurricane Harvey, and you might only focus on its heavy rainfall here.
*The sentence has been rewritten as: "Harvey brought a storm surge of up to 3 m and **widespread** torrential rain to the Texas coast, with the heaviest rainfall, over 1500 mm (60 in), measured at Nederland and Groves, near Houston (Blake and Zelinsky, 2018)."*

lines 73-94, same as previous comments.
*We think it is important to show by how much the flow in Texas rivers increased as a result of the rainfall from Harvey. However, we have removed the sentences beginning "Flow in the Trinity River..." (lines 84-86 in the original) and "Runoff can add nutrients...." (lines 89-92 in original).*

Methods:

lines 130-133, all the instrument should provide the company info.
*They do. Seabird Scientific, Chelsea Instruments, and Biospherical Instruments are all well-known manufacturers.*

line 155-156, please indicate whether the fluorometer data has been calibrated with in-situ Chl a data or not.

*We did not collect separate samples for chlorophyll-a, but the sensors were sent for calibration in December 2016 and again in March 2018. We have added a statement to this effect.*

Results:

line 153, You may want to consider to present wind field (not shown in this study) and current movement data together. It will then better show the impact of hurricane and other conditions on both variables.
*We believe this is unnecessary, given that the currents follow the wind pattern closely in this area (see later statement in manuscript).*

line 189, 191, Traditionally, reference(s) is not cited in the Results section.
*This is true, but it seemed easier to include the references here than try to extend the discussion section, where they would otherwise have to go.*

line 218, you may want to present the measured temperature as supplementary information.
*We do not believe this is necessary, as temperatures across the region sampled only varied by 1-2 degrees during each cruise.*

line 227, again, reference.
*See comment above*

line 299, reference again.
*See comment above*

lines 329-330, references again.
*See comment above*

Discussion.

This is a very long Discussion section, you may want to consider to separate it into different subtitles, e.g., impact on the Bay regions, inner shelf, offshore, phytoplankton, as well as oxygen dynamics?
*We have broken the discussion into five segments: the introductory paragraphs discuss what was expected from previous work on the effects of hurricanes on the coastal zone; then Oxygen and nutrient variability; Salinity variability in the coastal zone; Chlorophyll variability; and Why was there no hypoxia following Harvey?*

lines 551-552, Sabine Lake to Port Aransas Bay, the name should be marked or labeled in the map.
*We have added Port Aransas Bay to Fig. 2, while Lake Sabine is shown in Fig. 1.*

lines 589-590, please consistent with the unit of same variable throughout the ms, e.g., µg/L and mg m-3.
*All concentrations of chlorophyll-a have been converted to $mg\ m^{-3}$*

lines 600-603, so, the satellite-derived results might mislead the interpretation of your results?
*This is possible, especially given that the in situ sampling from the ship during the cruises showed only very low concentrations of chlorophyll-a.*

lines 608-618, I believe the statement regarding hypoxia formation has been well known, and you may want to simplify it here.

Lines 659-719, four factors have been proposed to explain why there was no hypoxia following Hurricane Harvey. The direct evidences of those factors should be provided to support your idea and persuade the reader.

Table 1, rates (cm hr-1)? If so, please revise it. Also, you should present the locations of precipitation in the sampling map. Please also consider move this table to supplementary information.

Fig. 1, You might want to mark and indicate the sampling area in this figure. The reader will have better idea where the stations located.
*Fig. 2 shows all stations sampled. We believe it is unnecessary to show them on Fig. 1 as well.*

Fig. 2, Please remove or describe the undescribed label in the figure. e.g., Yellow triangle X. Also, please mark the line(s) which mentioned in the text, e.g., line 3.
*We have added the identifier for TABS mooring X to the caption, along with a statement showing which stations are shown in other figures.*

Fig. 3, You might want to combined all panel (A, B, C, and D) of each buoy's data into a longer time period and also marked your pre- and post-hurricane sampling periods and hurricane period in the longer time panel's data plot. In this way, the reader can clearly picture all the time frame of your study.
*The problem with doing this is that it will make the figure even less readable than at present, even if it is turned sideways to use a landscape format (as we have now done) rather than a letter format.*

You may want to consider to combine Fig. 1-3 into one figure or at least Fig. 1 and 2. It will provide a clear picture for the reader regarding your study site, sampling period, and current condition of your study.
*We are not sure there is any advantage to doing this, and have left it as per the original.*

Fig. 4, it will be easier to follow your description if Mississippi-Atchafalaya river system, Galveston Bay, and Matagorda Bay are marked on the map.
*The positions of Galveston Bay and the Matagorda Bay system are already marked on Fig. 1. We have added the mouth of the Mississippi River (#4).*

Fig. 5, y-axis, the water column depth should mark with minus sign; Please mark pycnocline in DO profiles. Also, please clearly indicate which September cruise for this data set (I know which cruise, but it still should be clearly indicated in the caption).
*We are not sure why you want the depth shown as a negative number, since almost all oceanography papers (and atlases) assume that 0 marks the surface in vertical plots. We have changed the caption to show which September cruise is meant.*

Fig. S3, it is more make sense to me if you reverse x- and y-axes plot of this figure.
*We have reversed the axes for this figure.*

---

## Author Response (AR2)

Hoppema comments

Dates: please use format 24 August 2017 all through the manuscript (many cases)
*We have corrected all dates to this format*

L38 Please use format m s-1
*We have gone through the manuscript and used this format throughout*

Fig. 1: Delete a from 1a. What do the numbers mean and what are the units? Please define SW. Indicate Louisiana in the map.
*Done. The numbers are explained in the caption., as is the 1.5 SW (a rain gauge). We have added (in Louisiana) to the caption rather than try and add it to the map and clutter it further.*

L51-52 delete: more than 80 fatalities, and over $150 billion in economic damage (as this is no scientific knowledge)
*Done*

L68 Please add when the website was accessed
*Done throughout*

L111 delete: additional (as this is already included in "added")
*Done*

Methods section: Please provide precision and/or accuracy of all measurements. Data downloaded from websites: please add when the website was accessed.
*Done (and see comment above about websites)*

L176 (caption Fig. 3): delete the second "cruise"
*Done*

Fig. 3: I think the panels and current vectors and the labels are too small to be readily readable. Please try to enlarge it.
*This is now as large as I can get it on a single page. If this is still considered too small, we could split the graphs into two pages. Please let me know about this.*

L207 2°C (add C for Celsius)
*Done*

L250 Please do not use psu, since the salinity that you present is dimensionless. In

some cases you could also use the term practical salinity (to distinguish it from absolute salinity)
*We have removed all references to psu*

L258 Please delete 1.22 Mcfs, since these are unknown units and it suffices to present the SI units
*Done. However, in the discussion of the flow rates in the rivers, we have kept cfs (cubic feet per second) as well as $m^3s^{-1}$ as this is how the USGS records their data. I have added a line in the text to explain this.*

L288 and further: It is mentioned that oxygen is generally saturated, but no saturation values are given. Please add those to the concentrations.
*These have been added for 25°C and 30°C*

Fig. 7 caption L485: Use data formats like 4 September 2017
*Done*

L515 and further: sub-section chlorophyll: A large part of this section gives and describes data (which could thus be moved to the Results section) There is not much discussion in this section.
*We have moved most of this to the Results section (new section 3.5) as suggested. A short part is left in the Discussion.*

L568 Write full: Gulf of Mexico
*Done*

L856 If the journal consists of only one word, it should be written out full: Biogeochemistry
L864 idem ditto
*All references checked for this*

**Shiller comments**

The authors have satisfactorily revised their manuscript and I recommend publication. I do suggest a couple of minor technical corrections:

Fig. 1: The caption refers to 3 & 4, which are not shown on the image.
*We don't understand this, as numbers 1-4 are shown on the figure and explained in the caption.*

Line 265: should probably remove the word 'rates'.
*Done*

Figs. 5 and 6: The units shown on the images are 'umol/L' rather than µmol/L. In your spreadsheet, you can use Alt-230 (typed on the numeric keypad) to get a µ symbol. Alternatively, look in the ODV users' guide for how to type Greek letters.
*We have corrected these figures and any others to show µmol/L*

Fig 7: The image says 'µMol/L', which should be µmol/L.
*Corrected*